# ELIMINATING FALSE POSITIVES AND TRIVIAL POSITIVES IN CONTRASTIVE LEARNING

## ABSTRACT

Contrastive self-supervised learning critically depends on stochastic augmentations to generate positive pairs without quality-guaranteed mechanism. The potential low-quality positives, inclusive of false positives and trivial positives, hinder models from learning effective representations. To address these issues, we propose View Selection via 2-Fold Indicators (VS-2FI). It identifies the low-quality pairs of both types respectively via two indicators before eliminating them. On the one hand, in order to identify false positives, we introduce Semantic Consistency, and approximate it via the likelihood of two views co-occurring beyond chance. On the other hand, in order to identify trivial positives, we design Alignment Level, and estimate it by the minimum network depth required to align two views. VS-2FI discards view pairs that are either low in Semantic Consistency (potential false positives) or low in Alignment Level (potential trivial positives) to improve the overall quality of positive pairs. Extensive experiments elucidate the isolated and integrated effects of the two indicators, and demonstrate the consistent gains of VS-2FI across different contrastive learning frameworks.

## 1 INTRODUCTION

Modern computer-vision systems increasingly rely on self-supervised representation learning (SSL) to escape the bottleneck of large-scale manual annotation (Vincent et al., 2008; Komodakis & Gidaris, 2018; LeCun & Misra, 2021; He et al., 2022). Among the many SSL paradigms, contrastive learning (CL) has emerged as a cornerstone because of its conceptual simplicity and strong performance (Chen et al., 2020a; Zbontar et al., 2021; Assran et al., 2022; Siméoni et al., 2025). At its core, CL seeks representations that are invariant to data augmentations: by enforcing alignment between positive views (i.e., augmented views of the same image), the model learns to ignore nuisance variation (e.g., changes in scale, color, or lighting), while preserving meaningful semantics of image content (e.g., object categories) (Tian et al., 2020a; Misra & Maaten, 2020; Mitrovic et al., 2020).

In order to learn representations that faithfully capture semantics, the generated pairs should share the same semantics while differing in nuisance variables (Tian et al., 2020b). Guided by this principle, most CL methods carefully design their augmentation operators for generating separate views, and apply them twice to the same image to form a positive pair (Chen et al., 2020a; Grill et al., 2020). However, since each view of the positive pair is generated independently, the *inter-view* relationship between the two views is often overlooked, leading to two kinds of low-quality positives: *false positives* and *trivial positives*. *False positives* occur when the two views fail to share the same semantics. For example, as illustrated in Fig. 1b, the top row shows two views containing different objects (i.e., a cat and a dog), and the bottom row shows one view containing a dog while the other containing only background. Such false positives may lead the model to mistakenly associate features of semantically unrelated content (Peng et al., 2022). *Trivial positives*, in contrast, arise when the two views are near-duplicates without nuisance disparity. As shown in both rows of Fig. 1c, even though each view is heavily augmented, the augmentations coincide in such a way that the two views remain visually similar. During training, the model may exploit low-level cues, such as color histograms, to align them, failing to learn high-level semantic features (Chen et al., 2021a).

To tackle the issues, we design two novel indicators: *Semantic Consistency* and *Alignment Level*, that respectively identify false positives and trivial positives. On the one hand, Semantic Consistency quantifies how likely the two views share the same semantics. Accordingly, false positives that

|  (a) High-quality positives. | (b) False positives. | (c) Trivial positives. |

Figure 1: **Different types of positive pairs.** The types are labeled by our proposed VS-2FI.

are semantically different should exhibit low Semantic Consistency. On the other hand, Alignment Level quantifies the minimum level of features required to align the two views. Thus, trivial positives that share low-level cues should exhibit low Alignment Level. To make the indicators tractable, we define proxies to respectively estimate each indicator, where the computation of both proxies is realized via a dedicated estimator model.[1] To estimate Semantic Consistency, we leverage the insight that semantically inconsistent view pairs (e.g., a cat and a dog) are less likely to co-occur in the same image, and tend to appear in different images. Then, the proxy for Semantic Consistency is defined as posterior probability that two views co-occur, given their embedding similarity produced by the estimator. We theoretically guarantee that this proxy reflects the likelihood of two views co-occurring beyond chance by relating it to PMI (Church & Hanks, 1990). For Alignment Level, the proxy is defined as the minimum network depth of the estimator required to align the two views. This is motivated by a well-established understanding that representations at shallower layers capture features at lower levels of abstraction (Bau et al., 2017). Thus, this minimum network depth serves as a proxy for the minimum level of features required to align the two views (Alignment Level). Leveraging the two indicators, we develop *View Selection via 2-Fold Indicators (VS-2FI)*, which improves the overall quality of positive pairs by discarding the identified false and trivial positives.

Our contributions are summarized as follows:

- We design a novel framework, VS-2FI, which improves the quality of positive pairs in contrastive learning by identifying and eliminating both false and trivial positives.
- To identify false positives, we design Semantic Consistency, and estimate it via a proxy that is theoretically related to the likelihood of two views co-occurring beyond chance.
- To detect trivial positives, we introduce Alignment Level, and approximate it by the minimum network depth required to align two views.
- Extensive experiments analyze the respective and combined effects of the two indicators, and show that, by leveraging both indicators, VS-2FI consistently enhances representation quality across various contrastive learning frameworks.

## 2 RELATED WORK

**Contrastive learning.** Existing contrastive learning methods can be broadly categorized into three families based on their training objectives: instance-discrimination (Wu et al., 2018; Chen et al., 2021b), self-distillation (Grill et al., 2020; Chen & He, 2021), and feature decorrelation (Zbontar et al., 2021; Bardes et al., 2022). We provide a detailed overview of these objective families in Appx. F.1. Despite their differences in formulation, all these methods build on the same assumption: positive pairs share semantic content while differing in nuisance variables (Tian et al., 2020b). However, this is not strictly satisfied in practice, resulting in false positives and trivial positives. Our goal is to address these issues through an approach applicable to all three objective families.

**Methods to improve the quality of positives.** A prominent line of work tackles low-quality positives by explicitly intervening in the view generation pipeline. Some works rely on heuristic rules to control individual augmentation operators (e.g., cropping) to improve pair quality. Specifically,

---

[1]The estimator is trained via self-supervision without the need for labeled data.

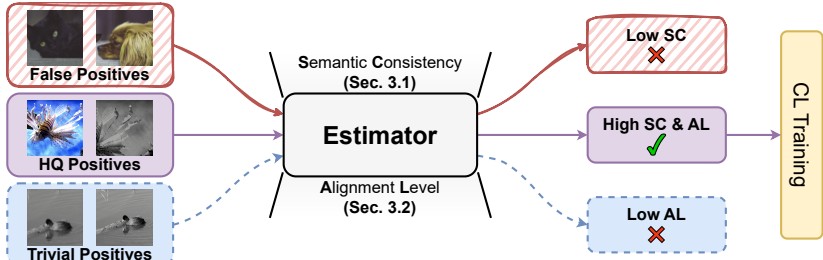

Figure 2: **The overall framework of VS-2FI.** For each positive pair, VS-2FI estimates its Semantic Consistency (Sec. 3.1) and Alignment Level (Sec. 3.2). It then discards pairs with low Semantic Consistency (potential false positives) or low Alignment Level (potential trivial positives), and selects pairs with high Semantic Consistency and high Alignment Level for CL training (Sec. 3.3).

Peng et al. (2022) employ saliency maps to constrain the cropping box around the object of interest, thereby suppressing false positives. Zhang et al. (2025) enforce a large area ratio between the two cropped views so that they retain a relatively stronger nuisance variation and are less likely to degenerate into trivial positives. However, these methods do not consider the interactions among various augmentations, limiting their effectiveness in addressing low-quality positives. In contrast, Ferreira et al. (2025) consider the interactions. Specifically, it generates multiple candidate views by the composition of multiple augmentations, and selects the hardest pairs–those with the highest sample-wise losses–as positives, thereby avoiding trivial positives. However, the sample-wise loss is not a proper indicator for two reasons. First, when models become capable, the hardest pairs are often false positives that cannot be easily aligned. Moreover, it is not applicable to CL objectives like feature decorrelation that cannot be decomposed into sample-wise losses.

**Sample selection in supervised learning.** In the context of supervised learning, many indicators have been proposed to select high-quality samples for training. For example, in active learning (Cohn et al., 1996; Settles, 2009; Gal et al., 2017; Sener & Savarese, 2018), uncertainty and diversity are commonly-used indicators to identify informative samples for annotation. In curriculum learning (Bengio et al., 2009; Kumar et al., 2010; Guo et al., 2018; Wang et al., 2021), both predefined and automatic difficulty indicators have been explored to identify easy and clean samples for initial training. In this paper, we study view selection in the context of contrastive self-supervised learning, and propose two novel indicators: Semantic Consistency and Alignment Level, to respectively identify false and trivial positives before eliminating them.

## 3 VS-2FI: VIEW SELECTION VIA 2-FOLD INDICATORS

To tackle false and trivial positives, we propose View Selection via 2-Fold Indicators (VS-2FI), as illustrated in Fig. 2. Concretely, it relies on two indicators: Semantic Consistency (discussed in Sec. 3.1) and Alignment Level (discussed in Sec. 3.2) to respectively identify false and trivial positives. In Sec. 3.3, we detail the view selection algorithm leveraging these two indicators to select high-quality positive pairs.

### 3.1 SEMANTIC CONSISTENCY

**Semantic Consistency as an indicator for false positives.** False positives arise when two views in the positive pair exhibit different semantics, as illustrated in Fig. 1b. To identify these false positives, Semantic Consistency quantifies how likely the two views share the same semantics, with low Semantic Consistency indicating potential false positives.

**The proxy to approximate Semantic Consistency.** Determining whether two views share the same semantics is fundamentally challenging without human annotations, as the semantic label of each view is unobservable. To tackle this challenge, we draw on the intuition that semantically inconsistent views (e.g., a cat and a dog) are *less likely* to co-occur within the same image, and

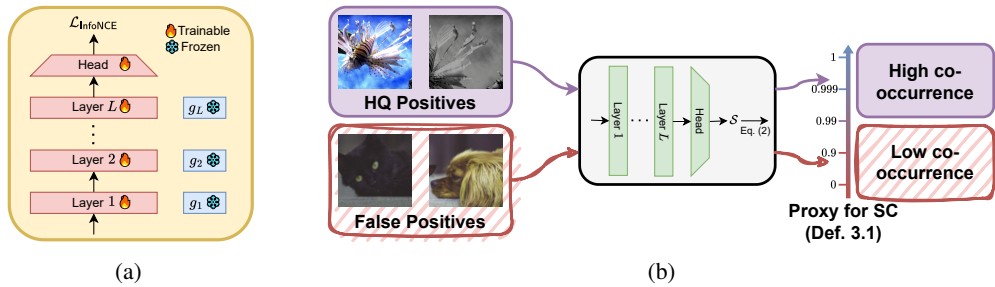

(a)                                                (b)

Figure 3: **Estimating Semantic Consistency.** (a) The estimator is trained with InfoNCE loss to estimate Semantic Consistency. (b) The proxy for Semantic Consistency reflects co-occurrence of two views; low co-occurrence implies low Semantic Consistency and potential false positives.

tend to appear more frequently in different images than semantically consistent ones.[2] This intuition underpins our approach: the proxy for Semantic Consistency should reflect the likelihood of two views co-occurring beyond chance. The estimation process is sketched in Fig. 3. Specifically, the estimation of Semantic Consistency requires an estimator model, which encodes each view $x$ into an embedding $z(x)$ and measures the co-occurrence likelihood of a pair $(x_1, x_2)$ by embedding similarity $\mathsf{sim}(z(x_1), z(x_2))$. To learn the embedding similarity, the estimator is trained to minimize InfoNCE loss (Oord et al., 2018):

$$\mathcal{L}_{\mathsf{InfoNCE}} = -\log \frac{e^{\mathsf{sim}(z(x_1^+), z(x_2^+))/\tau}}{\sum_{x' \in \{x_2^+\} \cup \mathcal{N}} e^{\mathsf{sim}(z(x_1^+), z(x'))/\tau}}, \tag{1}$$

where $(x_1^+, x_2^+)$ are two augmented views of the same image (a positive pair), $\mathcal{N}$ is a set of views drawn from different images, and $\tau$ is the temperature hyperparameter. The loss encourages the embedding similarity to be higher for co-occurring views than for views from different images, making the similarity act as a measure of co-occurrence. Thus, in order to make the proxy for Semantic Consistency reflect co-occurrence of two views, the proxy should be defined as a function of the embedding similarity. To formally establish the relationship between the proxy for Semantic Consistency and co-occurrence, we define the proxy as the posterior probability that two views co-occur, given their embedding similarity. Let pos denote the event that $(x_1, x_2)$ is jointly sampled from the same image, and neg the event that it is independently sampled from different images. The definition of the proxy is then given as follows:

**Definition 3.1** (Proxy for Semantic Consistency). Given two views $x_1$ and $x_2$ with similarity score $\mathcal{S} := \mathsf{sim}(z(x_1), z(x_2))$, the proxy for Semantic Consistency is defined as

$$\text{Semantic Consistency} \approx p(\mathsf{pos} \mid \mathcal{S}) = \frac{p(\mathcal{S} \mid \mathsf{pos})p(\mathsf{pos})}{p(\mathcal{S} \mid \mathsf{pos})p(\mathsf{pos}) + p(\mathcal{S} \mid \mathsf{neg})p(\mathsf{neg})}, \tag{2}$$

where $p(\mathcal{S} \mid \mathsf{pos}), p(\mathcal{S} \mid \mathsf{neg})$ denote the similarity distributions of positive and negative pairs, respectively, and $p(\mathsf{pos}), p(\mathsf{neg})$ are the prior probabilities of events pos and neg, respectively.

Operationally, the priors $p(\mathsf{pos})$ and $p(\mathsf{neg})$ are hyperparameters (set to 0.5 for simplicity), while $p(\mathcal{S} \mid \mathsf{pos})$ and $p(\mathcal{S} \mid \mathsf{neg})$ are estimated using histograms over the estimator's training data (visualized in Appx. B). Then the relationship between the proxy and co-occurrence can be established by connecting the proxy to pointwise mutual information (PMI) (Church & Hanks, 1990).

**Proposition 3.1** (Proxy for Semantic Consistency as sigmoid of PMI). *Let* $\mathcal{S}^{\star} = \mathsf{sim}(z^{\star}(x_1), z^{\star}(x_2))$ *be the similarity score produced by a converged encoder* $z^{\star}$ *under the InfoNCE loss. Then, we have*

$$p(\mathsf{pos} \mid \mathcal{S}^{\star}) = \frac{1}{1 + \frac{p(\mathsf{neg})}{p(\mathsf{pos})} e^{-\mathsf{PMI}(x_1, x_2)}}, \tag{3}$$

*where* $\mathsf{PMI}(x_1, x_2) = \log \frac{p(x_1, x_2)}{p(x_1)p(x_2)}$ *is the pointwise mutual information between views* $x_1$ *and* $x_2$.

---

[2]To clarify, semantically inconsistent view pairs are only less likely–not absent. Despite their relative rarity, we aim to eliminate such false positives during training to improve representation learning.

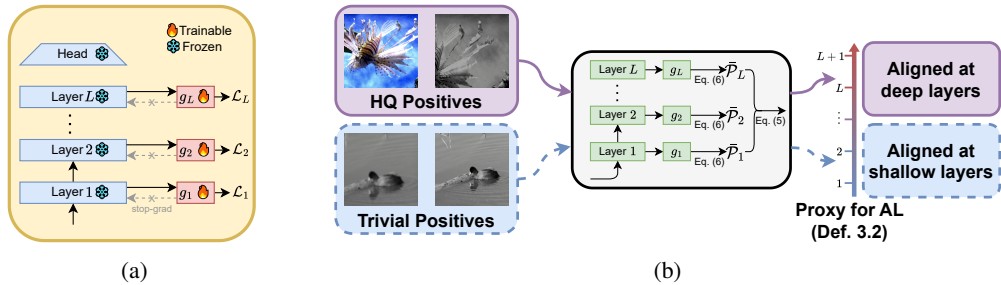

(a)                                                        (b)

Figure 4: **Estimating Alignment Level.** (a) The training of the early-exit projectors for estimating Alignment Level. (b) The proxy for Alignment Level is defined as the minimum depth required to align two views; low depth implies low Alignment Level and potential trivial positives.

The proof is presented in Appx. A. The PMI quantifies how likely two views co-occur beyond chance: a high PMI implies a high probability of co-occurrence $p(x_1, x_2)$ and a low probability of independent sampling $p(x_1)p(x_2)$. Thus, Prop. 3.1 indicates that the proxy likewise reflects the co-occurrence beyond chance.

## 3.2 ALIGNMENT LEVEL

**Alignment Level as an indicator for trivial positives.** Trivial positives occur when two views in the positive pair are nearly identical without significant variations in low-level features, as illustrated in Fig. 1c. To detect such trivial positives, Alignment Level quantifies the minimum level of features required to align the two views, with low Alignment Level indicating potential trivial positives.

**The proxy to approximate Alignment Level.** To estimate Alignment Level, we exploit the *hierarchical nature* of the features in neural networks (Zeiler & Fergus, 2014; Raghu et al., 2021): shallow layers encode low-level cues (e.g., color histograms), whereas deeper layers represent higher-level semantics (e.g., object categories). Accordingly, views with low Alignment Level should be aligned at shallow layers, bypassing the need for deeper semantic representations. In practice, we define the proxy for Alignment Level as the minimum depth of the *estimator* required to align the two views. The estimation process is illustrated in Fig. 4. Concretely, for each layer $\ell$ of the estimator, we attach an *early-exit projector* $g_\ell$ to the intermediate representation $h_\ell(x)$ at that layer, to extract features from it. These projectors are trained with InfoNCE loss:

$$
\mathcal{L}_\ell = -\log \frac{e^{\mathsf{sim}(z_\ell(x_1^+), z_\ell(x_2^+))/\tau}}{\sum_{x' \in \{x_2^+\} \cup \mathcal{N}} e^{\mathsf{sim}(z_\ell(x_1^+), z_\ell(x'))/\tau}}
$$

$$
=: -\log \frac{\mathcal{Q}_\ell(x_1^+, x_2^+)}{\mathcal{Q}_\ell(x_1^+, x_2^+) + \sum_{x^- \in \mathcal{N}} \mathcal{Q}_\ell(x_1^+, x^-)} =: -\log \mathcal{P}_\ell(x_1^+, x_2^+),
$$

(4)

where $z_\ell(x) := g_\ell(\mathsf{stop\text{-}grad}(h_\ell(x)))$, with stop-grad (Chen & He, 2021) which avoids interfering with the features encoded at layer $\ell$; $\mathcal{Q}_\ell(x_1, x_2) := e^{\mathsf{sim}(z_\ell(x_1), z_\ell(x_2))/\tau}$; and $\mathcal{P}_\ell(x_1, x_2)$ is referred to as the alignment score at layer $\ell$. By minimizing this loss (i.e., maximizing the alignment score $\mathcal{P}_\ell(x_1^+, x_2^+)$), each projector learns to align positive pairs by extracting their shared features from the corresponding layer's representations. Consequently, views with low Alignment Level should exhibit high alignment score $\mathcal{P}_\ell(x_1^+, x_2^+)$ at shallow layers. To operationalize this intuition, we define the proxy for Alignment Level based on the alignment scores of early-exit projectors:

**Definition 3.2** (Proxy for Alignment Level)**.** Let the network have $L$ layers indexed by $\ell \in \{1, \ldots, L\}$. The Alignment Level $\ell_{\mathsf{AL}}$ of a positive pair $(x_1^+, x_2^+)$ is defined as the minimum layer at which its alignment score exceeds a threshold:

$$
\text{Alignment Level} \approx \ell_{\mathsf{AL}} = \min\{\ell \mid \mathcal{P}_\ell(x_1^+, x_2^+) > \gamma\} \cup \{L + 1\},
$$

(5)

where $\gamma \in [0, 1)$ is the threshold hyperparameter.

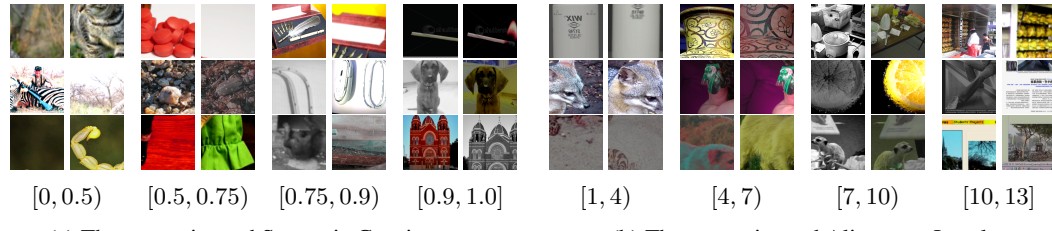

| $[0, 0.5)$ | $[0.5, 0.75)$ | $[0.75, 0.9)$ | $[0.9, 1.0]$ | | $[1, 4)$ | $[4, 7)$ | $[7, 10)$ | $[10, 13]$ |

(a) The approximated Semantic Consistency.   (b) The approximated Alignment Level.

Figure 5: **Examples of view pairs sampled from different ranges of both approximated indicators.** Each subfigure shows three pairs of views sampled from the corresponding indicator range. More examples can be found in Appx. D.

In implementation, we eliminate the randomness introduced by the sampling process of $\mathcal{N}$ in $\mathcal{P}_\ell(x_1^+, x_2^+)$ by marginalization:

$$\bar{\mathcal{P}}_\ell(x_1^+, x_2^+) = \mathbb{E}_{p(x_1'^+, \mathcal{N})} \frac{\mathcal{Q}_\ell(x_1^+, x_2^+)}{\mathcal{Q}_\ell(x_1^+, x_2^+) + \sum_{x^- \in \mathcal{N}} \mathcal{Q}_\ell(x_1'^+, x^-)}, \quad (6)$$

where the expectation is efficiently approximated by averaging over the histogram bins of $p(\sum_{x^- \in \mathcal{N}} \tilde{\mathcal{P}}_\ell(x_1^+, x^-))$, which is pre-computed on the estimator's training data (visualized in Appx. B).

### 3.3 THE VIEW SELECTION ALGORITHM

There are several potential ways to apply the two indicators within a CL pipeline, for example through view selection (Ferreira et al., 2025), loss re-weighting (Ren et al., 2018), or image synthesis (He et al., 2023). As a start, we leave the more complex methods for future work, and instead focus on the view selection method, which only requires a minor modification to the standard CL pipeline. The typical CL pipeline consists of two main stages. First, for each image $x$, it applies two compositions of augmentations $\mathcal{T}_1$ and $\mathcal{T}_2$ to generate two positive views $x_1^+ = \mathcal{T}_1(x)$ and $x_2^+ = \mathcal{T}_2(x)$. Second, it computes and optimizes the loss $\mathcal{L}_{\mathsf{CL}}(x_1^+, x_2^+)$. We leave the second stage unchanged, and slightly modify the first view generation stage. Specifically, given an image $x$, we repeatedly apply either augmentation $\mathcal{T}_1$ or $\mathcal{T}_2$ to generate a view set $\{x_i^+\}_{i=1}^M$, consisting of $M$ views. These views can be combined into $\binom{M}{2}$ candidate pairs. We then filter out pairs with low approximated Semantic Consistency ($p(\mathsf{pos}|\mathcal{S}) < \min\{\kappa_{\mathsf{SC}}, \max\{p(\mathsf{pos}|\mathcal{S})\}\}$) and low estimated Alignment Level ($\ell_{\mathsf{AL}} < \min\{\kappa_{\mathsf{AL}}, \max\{\ell_{\mathsf{AL}}\}\}$), where $\kappa_{\mathsf{SC}}$ and $\kappa_{\mathsf{AL}}$ are hyperparameters, and the min-max operation ensures that not all candidates are removed. Finally, we randomly sample a positive pair $(x_1'^+, x_2'^+)$ from the remaining candidates. The pseudocode of the view selection framework is presented in Appx. C.

## 4 EXPERIMENTS

### 4.1 IMPLEMENTATION DETAILS

**Training the estimator.** We train the estimator using the MoCo v3 framework with the InfoNCE objective. The estimator backbone is a ViT-S/16 with $L = 12$ transformer blocks. For each block at layer $\ell$, we attach a single-layer early-exit projector $g_\ell$ to its frozen representations stop-grad($h_\ell$), and train it with the same hyperparameters for computing loss and optimization as MoCo v3. We train two estimators separately: one on ImageNet-1K (IN1K) (Deng et al., 2009) and one on ImageNet-100 (IN100, a 100-class subset of ImageNet).

**Using the estimator.** We use the estimator trained on IN1K in most experiments, except for the CL pretraining experiments on IN100, where we use the estimator trained on IN100 to ensure a fair comparison. For estimating Semantic Consistency, we use the output embeddings of the estimators' head at the top layer to compute the embedding similarity $\mathcal{S}$ and the posterior probability $p(\mathsf{pos} \mid \mathcal{S})$ as defined in Eq. (2). For estimating Alignment Level, we set the per-layer probability threshold

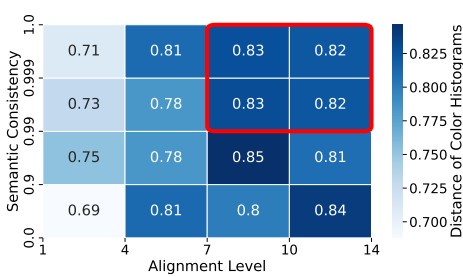 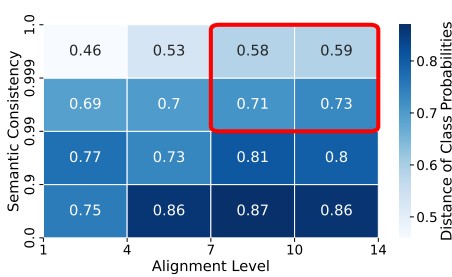

(a) Distance of color histograms (low-level).

(b) Distance of class probabilities (high-level).

Figure 6: **The relationship between approximated indicators and feature differences.** Feature differences are measured at both (a) low-level and (b) high-level. The red boxes indicate the selected view pairs with high Semantic Consistency ($\geq 0.99$) and high Alignment Level ($\geq 7$).

to $\gamma = 1/64$ and compute $\ell_{\text{AL}}$ by Eq. (5). For the histograms used in the approximation of both indicators, we set the number of bins to 20. During CL pretraining, we generate $M = 5$ candidate views for each image, and discard candidate pairs with $p(\text{pos} \mid \mathcal{S}) < \kappa_{\text{SC}} = 0.99$ (potential false positives) or with $\ell_{\text{AL}} < \kappa_{\text{AL}} = 7$ (potential trivial positives).

### 4.2 ANALYSIS OF THE INDICATORS' ISOLATED AND INTEGRATED EFFECTS

In this section, we analyze the isolated and integrated effects of the two proposed indicators by qualitative and quantitative experiments.

**Visualizing view pairs from different indicator ranges.** We first qualitatively analyze the isolated effects of the approximated Semantic Consistency and Alignment Level on identifying false positives and trivial positives. Fig. 5 summarizes view pairs sampled from different ranges of both indicators. In Fig. 5a, we observe that the sampled view pairs with low approximated Semantic Consistency ($< 0.75$) are often semantically different, which are considered as false positives, while pairs with high Semantic Consistency ($\geq 0.9$) are more semantically consistent. In Fig. 5b, we see that pairs with low Alignment Level ($< 7$) tend to be visually similar and thus are considered as trivial positives, while pairs with high Alignment Level ($\geq 7$) exhibit more substantial variations. These observations validate that the two indicators can do their intended jobs of identifying false positives and trivial positives, respectively.

**Quantifying feature distances at varying indicator values.** We further quantitatively analyze both isolated and integrated effects of the two indicators on selecting high-quality positive pairs. We divide the ranges of both approximated Semantic Consistency and Alignment Level into four intervals, resulting in a $4 \times 4$ grid of indicator value combinations. For view pairs in each cell of the grid, we compute their averaged low-level and high-level feature distances. For low-level features, we extract color histograms in the $L^*a^*b^*$ color space and compute their Hellinger distance. A small distance of color histograms indicates that the two views are visually similar, which is a sign of trivial positives. For high-level features, we use an image classifier (Steiner et al., 2022) to obtain class probability distributions and compute their Hellinger distance. A large distance of class probabilities indicates that the two views are semantically different, which is a sign of false positives. More details are supplied in Appx. E.1. Fig. 6 presents the results as heatmaps. First, we find that each indicator alone correlates well with the intended feature distance: Alignment Level positively correlates with low-level feature distance in Fig. 6a, while Semantic Consistency negatively correlates with high-level feature distance in Fig. 6b. This validates the isolated effects of both indicators. Moreover, the view pairs selected by VS-2FI (red boxes) tend to have large low-level feature distances but small high-level feature distances, indicating that VS-2FI reliably selects high-quality positive pairs with large low-level variations yet consistent semantics.

**Ablating the two indicators.** Previous qualitative and quantitative analyses have validated the effectiveness of the two indicators in identifying and discarding false positives and trivial positives.

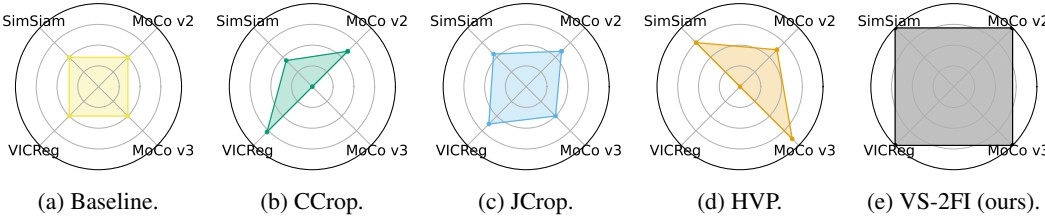

(a) Baseline.     (b) CCrop.     (c) JCrop.     (d) HVP.     (e) VS-2FI (ours).

Figure 7: **Benchmark of view selection methods.** The larger a polygon in a radar chart is, the better the corresponding view selection method performs across different objectives and backbones.

We further conduct an ablation study to assess their individual and combined contributions to CL performance. The results in Tab. 1 show that using either Semantic Consistency or Alignment Level alone to discard one type of low-quality positives improves performance over the baseline without view selection. Furthermore, using both indicators together to discard both types of low-quality positives achieves the best performance. This demonstrates the complementary benefits of the two indicators in improving CL.

Table 1: **Ablation study of indicators.** All models are pretrained on IN100 using MoCo v3 (100 epochs) and evaluated via linear probing.

| Semantic Consistency | Alignment Level | Accuracy | Δ |
|:---:|:---:|:---:|:---:|
| | | 75.2 | |
| ✓ | | 76.3 | +1.1 |
| | ✓ | 76.7 | +1.5 |
| ✓ | ✓ | 79.7 | +4.5 |

### 4.3 COMPARISONS WITH RECENT METHODS FOR IMPROVING VIEW QUALITY

**Experimental setup.** We benchmark recent methods for improving view quality, including CCrop (Peng et al., 2022), JCrop (Zhang et al., 2025), HVP (Ferreira et al., 2025), our proposed VS-2FI, and the baseline without view selection. These methods are evaluated across three typical CL objectives: instance-discrimination (MoCo v2 (Chen et al., 2020b), MoCo v3 (Chen et al., 2021b)), self-distillation (SimSiam (Chen & He, 2021)) and feature decorrelation (VICReg (Bardes et al., 2022)), and two popular backbones: ResNet-50 (He et al., 2016) (MoCo v2, SimSiam and VICReg) and ViT-S/16 (MoCo v3). All methods are pretrained on IN100 and evaluated via linear probing on IN100. More detailed experimental settings are supplemented in Appx. E.2.

**Results and analysis.** The results are summarized as radar charts in Fig. 7. VS-2FI consistently outperforms other methods on all objectives and backbones, demonstrating its effectiveness over prior methods that focus on individual augmentations (CCrop and JCrop) or individual types of low-quality positives (HVP). Moreover, one advantage of VS-2FI is its generality: it can be broadly applied to typical CL frameworks and backbone architectures, while CCrop is designed for CNN backbones and therefore is not applicable to MoCo v3, and HVP is only suitable for per-image losses and thus cannot be applied to VICReg.

### 4.4 PRETRAINING ON IMAGENET-1K

**Experimental setup.** We pretrain MoCo v3 and VICReg on IN1K with and without VS-2FI to evaluate its effectiveness on large-scale pretraining. The learned representations are then evaluated on the in-domain dataset (IN1K), out-of-domain datasets (C10 and C100 (Krizhevsky, 2009), Flwrs (Nilsback & Zisserman, 2008), Cars (Krause et al., 2013), DTD (Cimpoi et al., 2014), Aircraft (Maji et al., 2013)). For the pretraining of MoCo v3, we implement gradient accumulation to achieve an effective batch size of 4096, which results in the same performance as the original implementation but requires much less GPU memory. See Appx. E.3 for more experimental details.

**Results and analysis.** The results are shown in Tab. 2. Firstly, we observe that VS-2FI consistently improves the performance of both MoCo v3 and VICReg on IN1K, demonstrating its effectiveness in improving large-scale CL pretraining. Secondly, we observe that VS-2FI also improves perfor-

Table 2: **Evaluating VS-2FI on IN1K.** The learned representations are evaluated on both in-domain (IN1K) and out-of-domain datasets (C10, C100, Flwrs, Cars, DTD, Aircraft).

| Method | IN1K | C10 | C100 | Flwrs | Cars | DTD | Acft | Avg |
|---|---|---|---|---|---|---|---|---|
| MoCo v3 | 73.2 | 93.5 | 78.7 | 90.0 | 33.6 | 74.3 | 37.8 | 68.7 |
| +VS-2FI | **73.5** | **94.3** | **79.9** | **91.7** | **37.0** | **75.1** | **42.5** | **70.6** |
| VICReg | 70.8 | 89.1 | 70.8 | 90.1 | 40.6 | 75.9 | 43.1 | 68.6 |
| +VS-2FI | **71.5** | **89.5** | **70.9** | **91.7** | **44.5** | **76.5** | **45.6** | **70.0** |

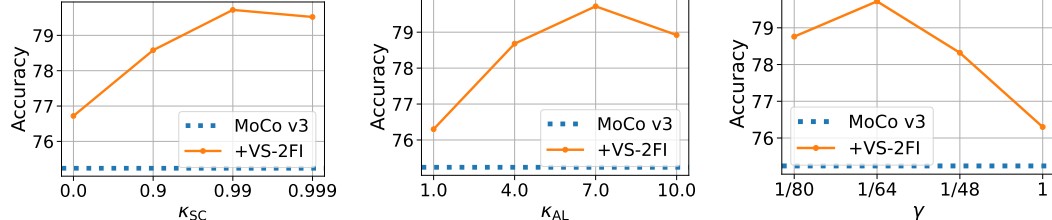

Figure 8: **Sensitivity analysis.** Models are pretrained on IN100 for 100 epochs.

mance across all out-of-domain datasets for both methods, indicating that the learned representations generalize better across domains.

### 4.5 STABILITY ANALYSIS

**Pretraining on non-object-centered and multi-object datasets.** Previous pretraining was conducted on ImageNet, which is featured by being single-object and object-centered. We further evaluate VS-2FI on less curated datasets characterized by being non-object-centered and containing multiple objects (MS-COCO (Lin et al., 2014)). Such datasets are more prone to false positives, since different augmented

Table 3: **Pretraining on MS-COCO.** The learned representations are evaluated on IN100.

| MoCo v3 | +VS-2FI | VICReg | +VS-2FI |
|---|---|---|---|
| 69.2 | **71.2** | 66.9 | **68.0** |

views of the same image may capture different objects with distinct semantics. We note that, with the same hyperparameters, the proportion of filtered pairs by Semantic Consistency increases from 0.08 on ImageNet to 0.12 on MS-COCO, reflecting the higher prevalence of false positives in multi-object scenes. As shown in Tab. 3, VS-2FI consistently improves over the baselines, highlighting its robustness on such datasets.

**Robustness to the choice of main hyperparameters.** We conduct sensitivity analysis on the main hyperparameters $\kappa_{SC}$, $\kappa_{AL}$ and $\gamma$. The results are shown in Fig. 8. VS-2FI achieves consistent improvements over the baseline across a reasonable range of hyperparameter values, demonstrating its robustness to the choice of hyperparameters.

## 5 CONCLUSION

In this paper, we address the issues of both false and trivial positives by a novel view selection framework, VS-2FI. It filters out low-quality positive pairs based on two indicators: Semantic Consistency and Alignment Level. Semantic Consistency quantifies how likely two views share the same semantics, and is approximated by the likelihood of two views co-occurring beyond chance. Alignment Level quantifies the minimum level of features required to align two views, and is approximated by the minimum network depth required to align them. Experimental results demonstrate the effectiveness of each indicator and the superiority of our VS-2FI framework over existing view selection methods across various contrastive learning frameworks.

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

APPENDIX

## A  PROOFS OF PROPOSITIONS

This appendix provides formal proofs for the Prop. 3.1. At the global minimizer of the expected InfoNCE loss, the optimal similarity function satisfies a density-ratio form, as originally shown by Oord et al. (2018):

**Lemma A.1** (InfoNCE optimality implies density-ratio). *The expected InfoNCE loss* $\mathbb{E}_{p(x_1^+, x_2^+, \mathcal{N})}[\mathcal{L}_{\mathsf{InfoNCE}}]$ *is minimized when the encoder* $z^\star$ *satisfies*

$$e^{\mathsf{sim}(z^\star(x_1), z^\star(x_2))/\tau} \propto \frac{p(x_1, x_2)}{p(x_1)p(x_2)} \quad \textit{for all } (x_1, x_2). \tag{7}$$

In other words, the exponentiated similarity score under the optimal encoder is proportional to an unnormalized density ratio between the joint distribution and the product of marginals.

*Proof of Lem. A.1.* We begin by rewriting the expected InfoNCE loss via a reparameterization. Originally, the loss is averaged over triplets $(x_1^+, x_2^+, \mathcal{N})$ drawn from the joint distribution $p(x_1^+, x_2^+, \mathcal{N})$.

We reparameterize the expectation as follows:

- Let $x_1^+$ be the anchor view.

- Construct a candidate list $V = [V_1, \ldots, V_N]$ that includes the positive view $x_2^+$ and negative views $\mathcal{N}$, randomly shuffled.

- Let $d \in \{1, \ldots, N\}$ denote the index of the positive view in $V$ (i.e., $V_d = x_2^+$).

Under this reparameterization, the expected InfoNCE loss becomes

$$\begin{aligned}
&\mathbb{E}_{p(x_1^+, x_2^+, \mathcal{N})}[\mathcal{L}_{\mathsf{InfoNCE}}] \\
&= \mathbb{E}_{p(x_1^+, V, d)}\left[-\log \hat{p}(d|V, x_1^+)\right] \\
&= \mathbb{E}_{p(x_1^+, V)}\left[\mathsf{CE}\left(p(d \mid V, x_1^+), \hat{p}(d|V, x_1^+)\right)\right],
\end{aligned} \tag{8}$$

where $\hat{p}(d \mid V, x_1^+) = \frac{e^{\mathsf{sim}(z(x_1^+), z(V_d))/\tau}}{\sum_{i=1}^N e^{\mathsf{sim}(z(x_1^+), z(V_i))/\tau}}$, and $\mathsf{CE}$ denotes the cross-entropy loss.

By standard results, the cross-entropy loss is minimized when the predicted distribution matches the true distribution:

$$\frac{e^{\mathsf{sim}(z(x_1^+), z(V_d))/\tau}}{\sum_{i=1}^N e^{\mathsf{sim}(z(x_1^+), z(V_i))/\tau}} = p(d \mid V, x_1^+). \tag{9}$$

To compute the RHS, we apply Bayes' rule assuming that negatives are sampled independently from $p(x)$:

$$\begin{aligned}
p(d \mid V, x_1^+) &= \frac{p(V_d \mid x_1^+) \prod_{j \neq d} p(V_j)}{\sum_{i=1}^N p(V_i \mid x_1^+) \prod_{j \neq i} p(V_j)} \\
&= \frac{\frac{p(V_d|x_1^+)}{p(V_d)}}{\sum_{i=1}^N \frac{p(V_i|x_1^+)}{p(V_i)}}.
\end{aligned} \tag{10}$$

Thus, the optimal score function satisfies

$$e^{\mathsf{sim}(z^\star(x_1), z^\star(x_2))/\tau} \propto \frac{p(x_2 \mid x_1)}{p(x_2)} = \frac{p(x_1, x_2)}{p(x_1)p(x_2)}. \tag{11}$$

This completes the proof. $\qquad\square$

Note that the form of density ratio in Lem. A.1 corresponds to pointwise mutual information (PMI) between the views $x_1$ and $x_2$:

$$\mathsf{PMI}(x_1, x_2) = \log \frac{p(x_1, x_2)}{p(x_1)p(x_2)}. \tag{12}$$

Therefore, the proportionality between the similarity score and the PMI can be expressed as:

$$e^{\mathsf{sim}(z^\star(x_1), z^\star(x_2))/\tau} = C \cdot e^{\mathsf{PMI}(x_1, x_2)}, \tag{13}$$

where $C > 0$ is an unknown proportionality constant independent of the particular pair $(x_1, x_2)$. Since $C$ is unknown and can vary across different training runs, the similarity score is not directly suitable for analysis. However, we can use the proxy for Semantic Consistency defined in Def. 3.1 to avoid this issue. This connection is formalized in Prop. 3.1.

*Proof of Prop. 3.1.* We begin by expressing the conditional distribution of the similarity score $s$ using the Dirac delta function:

$$p(s|x_1, x_2) := \delta\left(e^{s/\tau} - e^{\mathsf{sim}(z(x_1), z(x_2))/\tau}\right). \tag{14}$$

This implies that the similarity score is deterministic given a view pair $(x_1, x_2)$.

Let pos denote the event that $(x_1, x_2)$ is jointly sampled from the same image, and neg the event that it is independently sampled from all images. Then,

$$p(s|\mathsf{pos}) = \int\limits_{x_1, x_2} p(x_1, x_2)\delta\left(e^{s/\tau} - e^{\mathcal{S}/\tau}\right) dx_1 dx_2, \tag{15}$$

$$p(s|\mathsf{neg}) = \int\limits_{x_1, x_2} p(x_1)p(x_2)\delta\left(e^{s/\tau} - e^{\mathcal{S}/\tau}\right) dx_1 dx_2, \tag{16}$$

where $\mathcal{S} = \mathsf{sim}(z(x_1), z(x_2))$.

At convergence under the InfoNCE objective, Eq. (13) implies

$$\delta\left(e^{s/\tau} - e^{\mathcal{S}^\star/\tau}\right) = \delta\left(e^{s/\tau} - C\frac{p(x_1, x_2)}{p(x_1)p(x_2)}\right). \tag{17}$$

By the sifting property of the Dirac delta function, we then have

$$\begin{aligned}
&p(x_1, x_2)\delta\left(e^{s/\tau} - C\frac{p(x_1, x_2)}{p(x_1)p(x_2)}\right) \\
&= \frac{e^{s/\tau}}{C}p(x_1)p(x_2)\delta\left(e^{s/\tau} - C\frac{p(x_1, x_2)}{p(x_1)p(x_2)}\right).
\end{aligned} \tag{18}$$

Using this, we obtain the relationship between $p(s|\mathsf{pos})$ and $p(s|\mathsf{neg})$ as:

$$\begin{aligned}
&p(s|\mathsf{pos}) \\
&= \int\limits_{x_1, x_2} \frac{e^{s/\tau}}{C}p(x_1)p(x_2)\delta\left(e^{s/\tau} - C\frac{p(x_1, x_2)}{p(x_1)p(x_2)}\right) dx_1 dx_2 \\
&= \frac{e^{s/\tau}}{C}p(s|\mathsf{neg}).
\end{aligned} \tag{19}$$

Therefore, combining Eqs. (13) and (19), we obtain the relationship between their quotient and the PMI:

$$\frac{p(\mathcal{S}^\star|\mathsf{pos})}{p(\mathcal{S}^\star|\mathsf{neg})} = \frac{e^{\mathsf{sim}(z^\star(x_1), z^\star(x_2))/\tau}}{C} = e^{\mathsf{PMI}(x_1, x_2)}. \tag{20}$$

Finally, substituting into the definition of the proxy for Semantic Consistency in Eq. (2) yields

$$p(\mathsf{pos} \mid \mathcal{S}^\star) = \frac{1}{1 + \frac{p(\mathsf{neg})}{p(\mathsf{pos})}e^{-\mathsf{PMI}(x_1, x_2)}}. \tag{21}$$

This completes the proof. $\qquad\square$

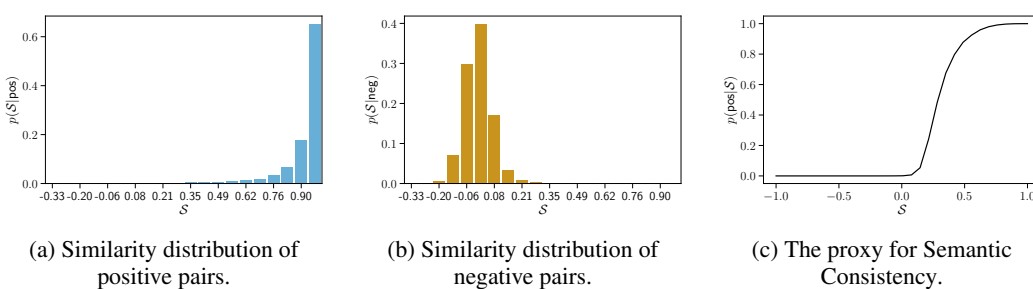

(a) Similarity distribution of positive pairs.

(b) Similarity distribution of negative pairs.

(c) The proxy for Semantic Consistency.

Figure 9: **Visualizing the distributions used to compute the proxy for Semantic Consistency.**

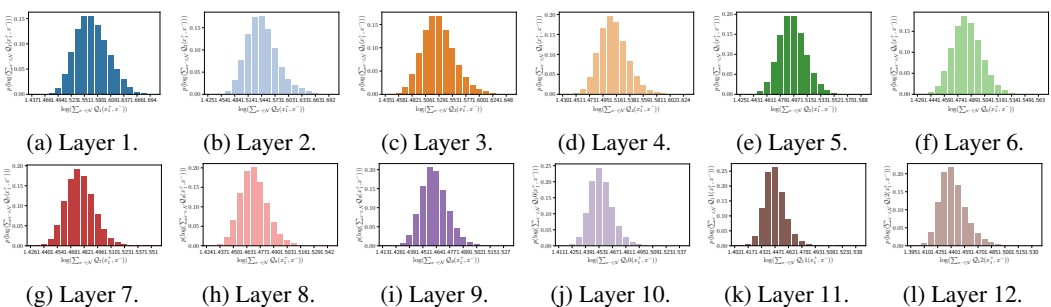

(a) Layer 1.    (b) Layer 2.    (c) Layer 3.    (d) Layer 4.    (e) Layer 5.    (f) Layer 6.

(g) Layer 7.    (h) Layer 8.    (i) Layer 9.    (j) Layer 10.    (k) Layer 11.    (l) Layer 12.

Figure 10: **Visualizing the distributions used to compute the proxy for Alignment Level.** Each subplot corresponds to one of the $\{p(\log(\sum_{x^- \in \mathcal{N}} \mathcal{Q}_\ell(x_1^+, x^-)))|\ell \in [1, 12]\}$.

## B VISUALIZATION OF HISTOGRAMS

In this section, we visualize the histograms used to compute the proxy for Semantic Consistency and the proxy for Alignment Level.

**Histograms for Semantic Consistency.** In Figs. 9a and 9b, we respectively visualize the histograms used to estimate the distributions $p(\mathcal{S} \mid \text{pos})$ and $p(\mathcal{S} \mid \text{neg})$ in Eq. (2). Based on these two distributions, we plot the Semantic Consistency curve in Fig. 9c, with values between bins interpolated linearly.

**Histograms for Alignment Level.** In Fig. 10, we present the histograms used to estimate the distributions $p(\log(\sum_{x^- \in \mathcal{N}} \mathcal{Q}_\ell(x_1^+, x^-)))$ in Eq. (6). To compute the expectation in Eq. (6), we approximate it by averaging over the histogram bins.

## C PSEUDOCODE OF VS-2FI

The PyTorch-like pseudocode of our VS-2FI method is shown in Alg. 1.

## D MORE EXAMPLES OF VIEWS WITH DIFFERENT INDICATOR VALUES

More examples of view pairs sampled from different ranges of both approximated indicators are presented in Fig. 11, complementing those in Fig. 5 in the main text. In Fig. 11, we observe similar trends as in Fig. 5. In particular, in Fig. 11a, we observe that the sampled view pairs with low approximated Semantic Consistency ($< 0.75$) are often semantically different, which are considered as false positives, while pairs with high Semantic Consistency ($\geq 0.9$) are more semantically consistent. In Fig. 11b, we see that pairs with low Alignment Level ($< 7$) tend to be visually similar and thus are considered as trivial positives, while pairs with high Alignment Level ($\geq 7$) exhibit more substantial variations. These observations validate that the two indicators can do their intended jobs of identifying false positives and trivial positives, respectively.

---

**Algorithm 1** Pseudocode of VS-2FI.

---

```
# T1, T2: two augmentation compositions
# L_CL: CL model and loss function
# estimator: estimator model
# M: number of candidate views
# k_SC: minimum Semantic Consistency threshold
# k_AL: minimum Alignment Level threshold

if VS_2FI: # enable view selection
    # generate candidate view pairs
    x_set = [choice([T1, T2])(x) for i in range(M)]
    candidates = pair(x_set) # num = M * (M - 1) / 2

    # estimate Semantic Consistency and Alignment Level
    SC, AL = estimator(candidates)

    # filter out candidates with low Semantic Consistency and Alignment Level
    filter(candidates, SC < min(k_SC, max(SC)))
    filter(candidates, AL < min(k_AL, max(AL)))

    # sample a pair from the remaining candidates
    x1, x2 = choice(candidates)
else:
    # standard view generation
    x1, x2 = T1(x), T2(x)

# compute and optimize the CL loss
loss = L_CL(x1, x2)
optimize(loss)
```

---

`choice`: uniformly choose one element; `pair`: generate all unique pairs.

---

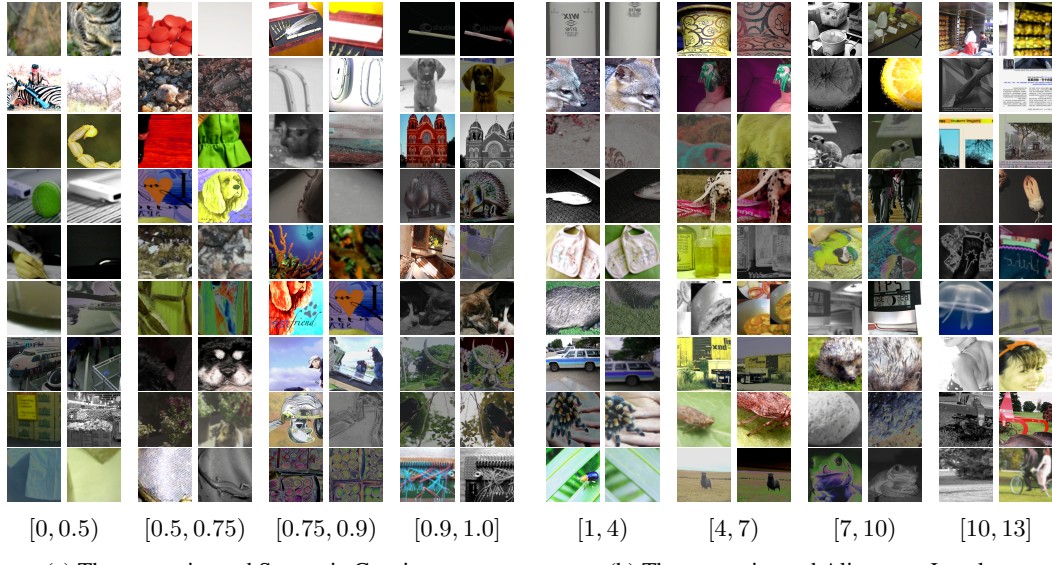

$[0, 0.5)$ $\quad$ $[0.5, 0.75)$ $\quad$ $[0.75, 0.9)$ $\quad$ $[0.9, 1.0]$ $\qquad$ $[1, 4)$ $\qquad$ $[4, 7)$ $\qquad$ $[7, 10)$ $\qquad$ $[10, 13]$

(a) The approximated Semantic Consistency. $\qquad$ (b) The approximated Alignment Level.

Figure 11: **More examples of view pairs sampled from different ranges of both approximated indicators.** Each subfigure shows nine pairs of views sampled from the corresponding indicator range.

# E    EXPERIMENTAL DETAILS

## E.1    EXPERIMENTAL DETAILS FOR SEC. 4.2

**Calculating distance of color histograms.**    For each view, we first convert the image to $L^*a^*b^*$ color space, and then compute the color histogram with 30 bins for each channel. Then for a pair of

views, we calculate the distance between their color histograms using the Hellinger distance:

$$d_{\text{Hellinger}}(H_1, H_2) = \sqrt{1 - \frac{1}{\sqrt{\bar{H}_1 \bar{H}_2 N^2}} \sum_I \sqrt{H_1(I) H_2(I)}}, \tag{22}$$

where $H_1$ and $H_2$ are the color histograms of the two views, $I$ is the bin index, $N$ is the total number of bins, and $\bar{H}_k = \frac{1}{N} \sum_I H_k(I)$ is the mean of histogram $H_k$.

**Calculating distance of class probabilities.** For each view, we feed it into a ViT-B/32 model pretrained on ImageNet-21K (Steiner et al., 2022), and obtain the predicted class probabilities over 21K classes. Then for a pair of views, we calculate the distance between their class probabilities using the Hellinger distance defined in Eq. (22).

E.2   EXPERIMENTAL DETAILS FOR SEC. 4.3

**CL frameworks.** For each framework (MoCo v2, SimSiam, VICReg and MoCo v3), we follow the official implementation and hyperparameters by default. We only modify the training epochs and the batch size to fit our computational resources:

| Hyperparameters | MoCo v2 | SimSiam | VICReg | MoCo v3 |
|---|---|---|---|---|
| Epochs | 200 | 200 | 200 | 300 |
| Batch size | 256 | 256 | 1024 | 1024 |

**Data augmentations.** Following BYOL (Grill et al., 2020), we use the strengthened augmentations for all methods, including random resized crop, horizontal flip, color jittering, random grayscale conversion, Gaussian blur and solarization. This strengthened augmentation brings significant performance improvement over the default augmentation used in MoCo v2 and SimSiam:

| Augmentations | MoCo v2 | SimSiam | VICReg | MoCo v3 |
|---|---|---|---|---|
| Not strengthened | 72.7 | 80.9 | - | - |
| Strengthened | 74.6 | 83.2 | 82.7 | 85.4 |

**Hyperparameters for other view selection methods.** For JCrop, we use the hyperparameters recommended in the original papers for each method. For HVP, we set the number of candidate views $M$ to 5, which is the same as our method, for a fair comparison. For CCrop, we tune the hyperparameters ($k = 0.4, \alpha = 0.6$) to achieve the best performance:

| Method | MoCo v2 | SimSiam | VICReg | MoCo v3 |
|---|---|---|---|---|
| Baseline | 74.6 | 83.2 | 82.7 | 85.4 |
| CCrop orig | 74.2 (-0.4) | 82.2 (-1.0) | 82.6 (-0.1) | - |
| CCrop tuned | 75.4 (+0.8) | 82.9 (-0.3) | 83.3 (+0.6) | - |

**Plotting radar charts.** We evaluate the learned representations via linear probing on ImageNet-100. The original accuracy values are

| Method | MoCo v2 | SimSiam | VICReg | MoCo v3 |
|---|---|---|---|---|
| Baseline | 74.6 | 83.2 | 82.7 | 85.4 |
| CCrop | 75.4 | 83.0 | 83.3 | - |
| JCrop | 75.4 | 83.4 | 83.0 | 85.4 |
| HVP | 75.6 | 84.1 | - | 86.1 |
| VS-2FI | **78.5** | **85.0** | **83.8** | **86.3** |

Then we normalize them to $[0, 1]$ for plotting radar charts in Fig. 7.

E.3   EXPERIMENTAL DETAILS FOR SEC. 4.4

For each CL framework, we use the official implementations and default hyperparameters unless otherwise noted. We make the following modifications to fit our computational resources. For VICReg, we reduce the training epochs to 200 and the batch size to 1024. For MoCo v3, we implement gradient accumulation with 4 steps to achieve an effective batch size of 4096, while

---

**Algorithm 2** MoCo v3 with gradient accumulation: PyTorch-like Pseudocode

---

```python
# f_q: encoder: backbone + proj mlp + pred mlp
# f_k: momentum encoder: backbone + proj mlp
# m: momentum coefficient
# tau: temperature
# grad_acc_steps: gradient accumulation steps

for x in loader: # load a minibatch x with N samples
    x1, x2 = aug(x), aug(x) # augmentation

    # split into grad_acc_steps chunks
    x1_steps, x2_steps = x1.chunk(grad_acc_steps), x2.chunk(grad_acc_steps)

    # compute keys for all samples
    k1_steps, k2_steps = [], []
    for i in range(grad_acc_steps): # gradient accumulation
        k1_step, k2_step = f_k(x1_steps[i]), f_k(x2_steps[i]) # keys: [N/grad_acc_steps, C
            ] each
        k1_steps.append(k1_step)
        k2_steps.append(k2_step)
    k1, k2 = cat(k1_steps), cat(k2_steps) # concat keys: [N, C] each

    # compute queries and loss, accumulate gradients
    for i in range(grad_acc_steps): # gradient accumulation
        q1_step, q2_step = f_q(x1_steps[i]), f_q(x2_steps[i]) # queries: [N/grad_acc_steps
            , C] each
        loss_step = (ctr(q1_step, k2) + ctr(q2_step, k1)) / grad_acc_steps # symmetrized
        loss_step.backward()

    update(f_q) # optimizer update: f_q
    f_k = m*f_k + (1-m)*f_q # momentum update: f_k

# contrastive loss
def ctr(q, k):
    logits = mm(q, k.t()) # [N, N] pairs
    labels = range(N) # positives are in diagonal
    loss = CrossEntropyLoss(logits/tau, labels)
    return 2 * tau * loss
```

---

`mm`: matrix multiplication; `k.t()`: k's transpose; `x.chunk(n)`: splits x into n chunks along the batch dimension; `cat`: concatenation. The prediction head is excluded from `f_k` (and thus the momentum update).

keeping the original training epochs (300) unchanged. We present the PyTorch-like pseudocode of the gradient accumulation in Alg. 2. This implementation results in the same performance as the original implementation with a batch size of 4096 (73.2 on IN1K), but requires much less GPU memory.

## F  RELATED WORK (FULL VERSION)

### F.1  CONTRASTIVE LEARNING

Contrastive learning (CL) has become a powerful paradigm for learning visual representations without human labeling (Balestriero et al., 2023; Gui et al., 2024). Its core mechanism is to learn representations that are invariant to nuisance variation while preserving semantics (Tian et al., 2020a; Misra & Maaten, 2020; Mitrovic et al., 2020). Several objective families have been proposed under this framework, with three prominent lines of work.

**Instance-discrimination.**  This class of methods treats each image instance as its own category, aiming to maximize similarity within the category (i.e., positive pairs from the same image) and minimize similarity across categories (i.e., negative pairs from different images) (Wu et al., 2018; Bachman et al., 2019; He et al., 2020; Chen et al., 2021b; Yeh et al., 2022).

**Self-distillation.**  These methods enforce invariance through a teacher-student framework, where a momentum-updated teacher encodes one view of a positive pair to produce a representation as the target, while the online student predicts this target from the alternate view of the positive pair (Grill et al., 2020; Chen & He, 2021; Caron et al., 2021; Oquab et al., 2024).

**Feature decorrelation.** Rather than maximizing agreement directly, this line of work constrains the cross-correlation matrix of positive-view representations to be close to the identity matrix (Zbontar et al., 2021; Bardes et al., 2022). The diagonal terms enforce invariance, while the off-diagonal terms reduce redundancy across feature dimensions.

Despite their differences in formulation, all three paradigms build on the same foundational assumption: positive pairs share semantic content while differing in nuisance variables (Tian et al., 2020b). However, this assumption is not strictly satisfied in practice. Standard view generation strategies produce each positive view independently, often resulting in two failure modes: *false positives*, which lack semantic consistency, and *trivial positives*, which exhibit little nuisance variation. Our goal is to address these issues through an approach applicable to all three objective families.

### F.2 METHODS TO IMPROVE THE QUALITY OF POSITIVES

Existing defenses against false positives and trivial positives fall into two main categories.

**More robust objectives.** These methods modify the training objective to make it more resilient to unreliable positive pairs. Robinson et al. (2021) alleviate trivial positives by introducing implicit feature modification to the InfoNCE loss to remove components of the current representations that are used to discriminate positive and negative pairs. Chuang et al. (2022) undermine the effect of false positives by introducing a robust loss that places more emphasis on easy positive pairs with low representation similarity. However, these methods are largely limited to InfoNCE-style losses and focus on reducing the impact of low-quality samples rather than addressing their root cause.

**Better view generation strategies.** A prominent line of work tackles low-quality positives by explicitly intervening in the view generation pipeline. Some methods employ stronger augmentation operators–such as color distortion (Chen et al., 2020a) or jigsaw transformations (Tian et al., 2020b)–disrupting shared low-level cues to reduce trivial positives. However, while these operators enhance the quality of individual views, the joint quality of the resulting positive pairs can still be sub-optimal. Some works rely on heuristic rules to control individual augmentation operators (e.g., cropping) to improve pair quality. Specifically, Peng et al. (2022) employ saliency maps to constrain the cropping box around the object of interest, thereby suppressing false positives. Zhang et al. (2025) enforce a large area ratio between the two cropped views so that they retain a relatively stronger nuisance variation and are less likely to degenerate into trivial positives. However, these methods do not consider the interactions among various augmentations, limiting their effectiveness in addressing low-quality positives. In contrast, Ferreira et al. (2025) consider the interactions. Specifically, it generates multiple candidate views by the composition of multiple augmentations, and selects the hardest pairs–those with the highest sample-wise losses–as positives, thereby avoiding trivial positives. However, the sample-wise loss is not a proper indicator for two reasons. First, when models become capable, the hardest pairs are often false positives that cannot be easily aligned. Moreover, it is not applicable to CL objectives like feature decorrelation that cannot be decomposed into sample-wise losses.

### F.3 SAMPLE SELECTION IN SUPERVISED LEARNING

In the context of supervised learning, many indicators have been proposed to select high-quality samples for training. For example, in active learning (Cohn et al., 1996; Settles, 2009; Gal et al., 2017; Sener & Savarese, 2018), uncertainty and diversity are commonly-used indicators to identify informative samples for annotation. In curriculum learning (Bengio et al., 2009; Kumar et al., 2010; Guo et al., 2018; Wang et al., 2021), both predefined and automatic difficulty indicators have been explored to identify easy and clean samples for initial training. In this paper, we study view selection in the context of contrastive self-supervised learning, and propose two novel indicators: Semantic Consistency and Alignment Level, to respectively identify false and trivial positives before eliminating them.

