# OpenReview forum: "Eliminating False Positives and Trivial Positives in Contrastive Learning"
_ICLR.cc/2026/Conference — ICLR 2026 Conference Withdrawn Submission_

### Official Review · Reviewer_UW6V · 2025-10-25

**Soundness:** 2
**Presentation:** 2
**Contribution:** 2
**Rating:** 4
**Confidence:** 4

**Summary:**

This paper addresses a fundamental issue in contrastive self-supervised learning (CL): the generation of low-quality positive pairs. The authors correctly identify two specific failure modes: false positives (views that do not share semantic content) and trivial positives (views that are too similar, lacking meaningful nuisance variation). To mitigate these, the paper proposes View Selection via 2-Fold Indicators (VS-2FI), a method that filters out such pairs before they are used for training. VS-2FI employs two novel, independently estimated indicators:

**Strengths:**

1. The paper tackles a clearly defined and widely recognized problem in CL. The distinction between false and trivial positives is precise and provides a useful framework for analyzing view quality.

2. The two proposed indicators are novel and well-grounded. The link between Semantic Consistency and PMI via the InfoNCE objective is theoretically sound. The use of early-exit projectors to estimate Alignment Level is a clever and intuitive way to leverage the hierarchical nature of neural networks.

**Weaknesses:**

1. The method introduces significant additional complexity. It requires pre-training a separate estimator model and generating multiple candidate views for each image. While the performance gains are compelling, a more detailed discussion of the computational cost (e.g., wall-clock time, memory usage) compared to baselines would be valuable for practitioners.

2. While sensitivity analysis is provided (Figure 8), the need to tune these parameters could be a barrier to adoption. Exploring more adaptive thresholding schemes could be a fruitful direction for future work.

3. The entire VS-2FI pipeline relies on the quality of the pre-trained estimator. The paper assumes the estimator is trained on the same or a similar domain. The robustness of the method when the estimator is trained on a distribution different from the target task is not explored.

**Questions:**

See weaknesses.

---

### Official Review · Reviewer_pte8 · 2025-10-26

**Soundness:** 3
**Presentation:** 3
**Contribution:** 3
**Rating:** 4
**Confidence:** 4

**Summary:**

I think this work is trying to tackle the presence of low-quality positive pairs generated by stochastic augmentations which may be a problem in self-supervised contrastive learning. The authors categorize these into two types: false positives, where the two augmented views fail to share the same semantic content (e.g., cropping two different objects from one image), and trivial positives, where the views are near-duplicates and lack meaningful nuisance variation.

They propose View Selection via 2-Fold Indicators (VS-2FI) which is a filtering mechanism that identifies and removes these bad pairs before they are used in the contrastive learning loss. The framework relies on two metrics calculated by a dedicated "estimator" model: 1) Semantic Consistency (SC): This one is designed to catch false positives. It quantifies how likely two views are to share the same semantics, which is approximated by their probability of co-occurring. 2) Alignment Level (AL): This one targets trivial positives. It's cleverly defined as the minimum network depth of the estimator that is required to align the two views. The core idea is that trivial pairs can be aligned using only shallow, low-level features. Then they conduct several experiments to show that this framework improves performance across various contrastive learning methods (like MoCo v2/v3, SimSiam, and VICReg) and datasets (including ImageNet and MS-COCO).

**Strengths:**

So I think this paper the problem is clearly articulated. The explicit separation of low-quality pairs into "false positives" and "trivial positives" is a valuable contribution on its own, as it helps clarify the two distinct failure modes of augmentation.

Also, I think the empirical validation is good to show. The qualitative examples in Figure 5 and the quantitative heatmaps in Figure 6 provide some evidence that the indicators are working as intended. Figure 6 shows that the selected pairs (high SC, high AL) have large low-level feature distances (not trivial) but small high-level feature distances (semantically consistent). The ablation study in Table 1 also shows that the two indicators are complementary and that using both gives the best results.

The authors also show that VS-2FI is more general than prior work. For instance, they note it can be applied to feature decorrelation methods like VICReg, whereas other methods like HVP are not suitable. The fact that it also performs well on MS-COCO, which is more prone to false positives than the object-centered ImageNet, which is a strong point in its favor.

**Weaknesses:**

I am curious about the computational overhead of this method, which is not really discussed. This method introduces a component: a pre-trained estimator model, which itself is a ViT-S. On top of that, the data loading pipeline for the actual CL training is made much more complex. Instead of just generating two views, the model must generate M=5 candidate views for each image to create 10 pairs. Then, in my opinion, it must run all 10 pairs through the estimator to get their SC and AL scores before a single training step can even be performed. This seems like it would dramatically slow down training. I want to see how this paper deal with a completed analysis of this overhead compare with the baseline.

My second concern is the generalizability on this estimator model. The approach hinges on having a good, well-calibrated estimator. The authors train their estimators on the target datasets (IN100 for IN100 experiments, IN1K for IN1K experiments). But I am just curious how would this work in practice on a new, uncurated dataset? Would one have to train an estimator first on that same dataset? It's not clear how sensitive the method is to the estimator's quality or if a generic, off-the-shelf estimator (like one trained on IN1K) would work for pre-training on a completely different domain (I guess the authors could try with medical images or satellite data). Also, I only saw MoCo v3 and VICReg's results in the paper's main experiment, I think it would be better to show the method is more generalize to other methods (e.g., BYOL, SimCLR, DCL..).

**Questions:**

My primary concerns are the unaddressed practical limitations. The method introduces a heavy ViT-S estimator and a complex sampling pipeline, which implies a massive computational overhead that is never analyzed. Furthermore, the approach's generalizability is questionable. The estimator is trained on the target dataset (e.g., IN1K for IN1K experiments) , and it is unclear if this is a practical requirement for new datasets or if a generic estimator would even work. Finally, the main experiments are limited to MoCo v3 and VICReg , failing to demonstrate that the method is broadly applicable to other key CL families like BYOL or SimCLR.

---

### Official Review · Reviewer_rcKk · 2025-10-29

**Soundness:** 2
**Presentation:** 4
**Contribution:** 3
**Rating:** 4
**Confidence:** 4

**Summary:**

This paper tackles the question that, in contrastive learning, the original views may suffer from bad augmentations. Specifically false positives and trivial positives. The research idea naturally convinces. It proposed two metrics to filter high-quality pairs for training and the experiment shows the effectiveness. However, from my perspective, there could be much more experiments to make this work more solid and for the community to reproduce.

My evaluation at this point is a weak rejection, given a lot of missing experiments from my perspective to verify the effectiveness of VS-2FI in a variety of ablation/case study settings to improve this paper.

**Strengths:**

The research problem is interesting, and the intuition appears sound to me. It is naturally possible that, augmentations are usually random, and may introduce bad augmentations. This work improves self-supervised learning in a data-centric manner by improving the data being used. Though it sounds a bit engineering, I personally welcome engineering approaches.

The part of aligning at deep layers and aligning at shallow layers is insightful from my perspective. It could be associated with some interpretability direction (i.e., why a pair is identified as good alignment or not). I also like Figures 5 and 11, presenting examples of proposed metrics (but I find the explanations lacking; see weaknesses).

Experiments are ok to me, but could be better. This paper could get sota results for the main task yet a lot of ablation and details are missing.

**Weaknesses:**

W1. Beyond just presenting the pairs and the numbers, there should be accompanying explanations of why some pairs have a larger semantic consistency/alignment level while other pairs do not.

W2. The hyperparameters (the threshold) highly affect the experiment results, and there should be more ablation with them. In the current manuscript there are only 4 choices for both K_{SC} and K_{AL}

W3. There should be a scaling experiment (using different sizes of dataset, maybe sampling sub-dataset) for pretraining on ImageNet-1K. There is an experiment on ImageNet-100 but it is not enough from my perspective.

W4. A lot of experiment details are missing (or maybe I missed while reading). For example, the hardware and total compute used in the experiment, how much memory and time it takes to conduct training, what's the choice for each hyperparameter (beyond batch size) in the experiment, etc. I don't see supplementary materials/codes provided, so these are necessary for a paper to be published in ICLR.

W5. It should be reported what ratio of augmented pairs are filtered/kept after the view selection. Following these, if we set a harsher threshold for a pair to be "high quality", will the performance improve because you're using better data?

W6. At the cost of selection, there will be time cost and resample cost, which should all be reported.

W7. In-depth case studies of why some pairs are marked as high quality, why some pairs are marked as false positives, and why some pairs are marked as trivial positives are needed to justify that the trained estimator is functioning well.

W8. How would the proposed VS-2FI compare with non-contrastive-learning methods, such as supervised training of neural networks on the given labeled data? An extention to this question is how the proposed VS-2FI compared with contrastive learning methods that are not centered on view selection?

W9. Is VS-2FI robust to the choice of different contrastive losses?

**Questions:**

I note some questions while reading this paper from the beginning to the end. Please feel free to address these in whatever order the author thinks the best.

Q1. I don't quite get why Figure 1(b) view 1 is a false positive? what are the labels for the original views (which should be included in the figure)

Q2. In figure 1, the pair (a original, a view1) and (c original, c view1) is similar (the view 1 is augmented with symmetric, crop and gray-scale). Why is one high-quality positives but the other trivial positives?

Q3. Just a random thought: in Figure 3, the training of estimator is using InfoNCE which is contraistive learning. That being said, is it possible to iteratively apply the proposed contrastive learning method to train a better estimator, and then use the better estimator ("second-order improvement") to train the actual self-supervised learning (first-order improvement)? This work right now focus on first-order improvement. Please let me know if I was understanding wrongly.

Q4. In equation 1, why the first sign is approximate? (if defining a new term shouldn't we use ":="?)

Q5. I know contrastive learning, but I'm just not an expert of view selection direction. Could the authors provide more related works? (There are two 2025 works in "Methods to improve the quality of positives" in section 2 and appendix F.2., which I appreciated. But what is the research landscape from 2022 to 2025?)

Q6. Is it possible for the proposed methods to be applied to domains other than computer vision?

---

### Official Review · Reviewer_ivnS · 2025-11-01

**Soundness:** 3
**Presentation:** 2
**Contribution:** 2
**Rating:** 4
**Confidence:** 4

**Summary:**

This paper addresses the problem of low-quality positive pairs in contrastive self-supervised learning, which arise from standard stochastic augmentation. The authors categorize these into false positives (semantically different views) and trivial positives (near-duplicate views). They propose a framework, VS-2FI, which uses Semantic Consistency and Alignment Level to identify false and trivial positives. The main contrastive model is trained only on the remaining high-quality pairs, improving the performance.

**Strengths:**

1. The paper is well motivated and well written.
2. The connection between semantic consistency and MI is clever.
3. The idea behind alignment level is also nice and the fact that authors use the same model for both semantic consistency and alignment level helps with the additional compute and parameter size added to the framework.

**Weaknesses:**

While the paper introduced a novel approach for removing low-quality positives, it has the following weaknesses:

1. While authors have presented a large set of experiments for comparison of their approach across multiple datasets, they have neglected some critical baselines. Particularly, the followings:

1.1. Comparison to SimCLR: The two metrics proposed in this paper are trained using an InfoNCE-style loss, which is the core objective of SimCLR. This makes SimCLR a uniquely important baseline for the main CL model. It is unclear how VS-2FI would interact with a SimCLR main model. Would the filtering provide orthogonal benefits, or would the estimator simply be re-learning the main model's objective, leading to diminishing returns? An experiment applying VS-2FI to SimCLR would provide much deeper insight into the method's mechanics and generality.

1.2 Comparison to multiple positive view: To select the high-quality data, the authors generate multiple positive samples and then filter them according to their proposed metrics for final positive pair. There is a line of work of having multiple positives, or poly-view contrastive learning, that systematically increase the number of positives and show the improvement of representations. This baseline seems quite necessary as it proposes an easier solution of generating samples and using all of them in the final loss without any hyperparameter tuning like the level of alignment or any filtering. It would be insightful and more fair if the authors also compare their result with this baseline to show that their approach is still improving or providing competitive results.

2. The method introduces additional computational and memory overhead by requiring: (a) the training of a separate estimator network, and (b) a generate-and-filter step within each training iteration of the main model. The paper does not quantify this overhead, making it difficult to assess the method's practical viability.
More importantly, the evaluation is not performed on a fair computational basis. A more rigorous comparison would be compute-matched: does VS-2FI outperform a standard baseline (e.g., MoCo v3) that is simply trained for more epochs using the same total computational budget? Without this analysis, it is hard to know if the proposed method is a more efficient use of computational resources than the simpler alternative of just training longer.

**Questions:**

1. The Alignment Level is defined as the minimum depth at which the alignment score exceeds a threshold $\gamma$. Authors mention $\gamma = 1/64$. Was this value tuned as a hyperparameter or what logic was behind choosing it? How sensitive the result is to these choices?
2. Have you considered alternatives to hard-threshold filtering? For example, could the Semantic Consistency and Alignment Level scores be used to re-weight the loss for each positive pair, providing a softer curriculum? This might be more robust and avoid the need for the min-max heuristic.

---

### Official Review · Reviewer_x97n · 2025-11-03

**Soundness:** 3
**Presentation:** 3
**Contribution:** 2
**Rating:** 2
**Confidence:** 4

**Summary:**

This paper aims to improve contrastive learning by eliminating potential low-quality positives, namely false positives and trivial positives from stochastic augmentations. To achieve this, the authors propose VS-2FI (View Selection via 2-Fold Indicators), which calculates Semantic Consistency for identifying false positives and Alignment Level for identifying trivial positives. Both indicators are estimated from outputs of a trained estimator, which is also trained by contrastive learning (InforNCE loss). Qualitative and quantitative experimental results demonstrate the effectiveness of VS-2FI in finding false positives and trivial positives, and improving constrastive learning.

**Strengths:**

- False positives and trivial positives generated by stochastic augmentations are critical problems for contrastive learning. The proposed VS-2FI approach is intuitive and effective.
- The experimental results are sufficient, and the qualitative and quantitative results are impressive.

**Weaknesses:**

- Although effective, the innovation of the proposed VS-2FI method is relatively limited. The overall idea of VS-2FI is largely consistent with commonly used data cleaning tricks—namely, training a target model on the initial dataset and using this model to filter data in order to construct a refined training set. Specifically, this baseline approach identifies low-score outputs of the initial target model as false positives and extremely high-score outputs as trivial positives, which is very similar in intention and effect to the proposed Semantic Consistency and Alignment Level concepts. Likewise, the proposed Semantic Consistency and Alignment Level also require a pre-trained estimator, which is trained through multi-view contrastive learning. The paper needs to further clarify the significant differences between the proposed method and this baseline, and provide quantitative comparisons to highlight its distinctive advantages and contributions.
- The proposed VS-2FI method requires pre-training an estimator, which itself is trained via contrastive learning on noisy dataset (i.e., false positives and trivial positives). This implies that the estimator’s outputs may be inaccurate, thereby limiting the potential upper bound of improvement achievable by the VS-2FI method. This limitation is non-negligible, and providing a more quantitative analysis of its impact would constitute an important contribution.

**Questions:**

See Weaknesses

---

### Note · Authors · 2025-11-13

I have read and agree with the venue's withdrawal policy on behalf of myself and my co-authors.